# Optical Analysis of Perovskite III-V Nanowires Interpenetrated Tandem Solar Cells

**DOI:** 10.3390/nano14060518

**Published:** 2024-03-14

**Authors:** Matteo Tirrito, Phillip Manley, Christiane Becker, Eva Unger, Magnus T. Borgström

**Affiliations:** 1NanoLund and Division of Solid State Physics, Lund University, Box 118, 221 00 Lund, Sweden; matteo.tirrito@ftf.lth.se; 2JCMwave GmbH, Bolivarallee 22, 14050 Berlin, Germany; phillip.manley@jcmwave.com; 3Solar Energy Division, Helmholtz-Zentrum Berlin für Materialien und Energie GmbH, 12489 Berlin, Germany; christiane.becker@helmholtz-berlin.de (C.B.); eva.unger@helmholtz-berlin.de (E.U.)

**Keywords:** photovoltaics, nanowires, perovskite, multi-junctions, optical modeling

## Abstract

Multi-junction photovoltaics approaches are being explored to mitigate thermalization losses that occur in the absorption of high-energy photons. However, the design of tandem cells faces challenges such as light reflection and parasitic absorption. Nanostructures have emerged as promising solutions due to their anti-reflection properties, which enhances light absorption. III-V nanowires (NWs) solar cells can achieve strong power conversion efficiencies, offering the advantage of potentially integrating tunnel diodes within the same fabrication process. Metal halide perovskites (MHPs) have gained attention for their optoelectronic attributes and cost-effectiveness. Notably, both material classes allow for tunable bandgaps. This study explores the integration of MHPs with III-V NWs solar cells in both two-terminal and three-terminal configurations. Our primary focus lies in the optical analysis of a tandem design using III-V semiconductor nanowire arrays in combination with perovskites, highlighting their potential for tandem applications. The space offered by the compact footprint of NW arrays is used in an interpenetrated tandem structure. We systematically optimize the bottom cell, addressing reflectivity and parasitic absorption, and extend to a full tandem structure, considering experimentally feasible thicknesses. Simulation of a three-terminal structure highlights a potential increase in efficiency, decoupling the operating points of the subcells. The two-terminal analysis underscores the benefits of nanowires in reducing reflection and achieving a higher matched current between the top and the bottom cells. This research provides significant insights into NW tandem solar cell optics, enhancing our understanding of their potential to improve photovoltaic performance.

## 1. Introduction

The photovoltaic industry is approaching the theoretical power conversion efficiency for single bandgap solar cells. The efficiency reachable due to the fundamental losses by a single-junction solar cell is known as the Schockley–Queisser limit [1]. Photons with energy higher than the bandgap result in higher energy carriers; this extra energy is dissipated in heat, a phenomenon known as thermalization loss. Multi-junction device structures address thermalization losses, introducing more absorbers.

Light reflection is among the problems to tackle in designing solar cells. It arises from the refractive index mismatch between the device and the air, and it can be reduced by adjusting the refractive index and film thickness. Anti-reflection coatings are commonly employed, but they are effective only for small wavelength ranges. Alternatively, scattering structures can enhance the optical path length, but they may introduce parasitic absorption or electrical defects. Prior research has demonstrated the anti-reflection properties of nanowire arrays (NWs) [2,3], resulting in enhanced light absorption due to optical modes forming in NWs [4,5], as well as their geometrical dependence [6].

Despite covering approximately 10% of the entire surface, III-V NWs have demonstrated strong power conversion efficiency [7,8], making them a cost-effective and material-saving solution for future applications.

The development of new lithography techniques, such as nanoimprint and Talbot displacement lithography, has contributed to the interest in nanostructures. Nanoscale structures have a relatively large surface-to-volume ratio, which allows strain relaxation at the surface [9].

In the last decade, metal halide perovskites (MHPs) have attracted growing interest in the field of photovoltaics due to their impressive optoelectronic properties, such as defect tolerance [10] and bandgap tunability [11,12]. MHPs are also known for their low production cost and reduced process complexity [13]. However, challenges related to the stability [14] and hysteresis [15] are observed for perovskite solar cells. Many reports have investigated the integration of MHPs with Si solar cells due to the commercial availability of the latter technology [16,17,18]. These studies have shown promising results in terms of power conversion efficiency for planar structures [19,20] and different tandem architectures [21,22]. Raja et al. focused on the integration of perovskite nanostructures with Si [23], obtaining a 21% power conversion efficiency enhancement with respect to the planar counterpart. In this paper, we investigate from an optical perspective a tandem structure using III-V, direct bandgap semiconductor NW arrays in tandem with perovskite. Both materials allow for bandgap tunability in the range of 1.2 eV to 2.4 eV [11], making them ideal for use in a tandem structure. As reported by Jäger et al. [24], the maximum power conversion efficiency of four-terminal and two-terminal tandems can be achieved if bottom cells bandgap is ∼1.0 eV.

The small footprint of NWs arrays leaves potential space for light harvesting. Thus, this space can be filled with another absorber, leading to an interpenetrated tandem structure. At first, we model the bottom cell and then the full tandem structure, varying the layer thicknesses to maximize absorption. In this work, an interpenetrated tandem solar cell is investigated from the optical perspective, the modeled structures are based on InP NWs solar cells developed at Lund NanoLab, taking into account experimentally available solution processing methods such as inkjet printing [25] and spin-on-patterning [26]. This structure implements a facile design that does not require anti-reflection coatings or patterned contacts, with minimal material consumption. We aim to realize a first template which can be further developed with different materials. We focus on three-terminal and two-terminal structures, obtaining a similar power conversion efficiency to Raja et al. [23]; however, the processability of the latter is more challenging, involving perovskite NWs, while the four-terminal structure has a more complex connection scheme.

## 2. Materials and Methods

### 2.1. Layout and Materials

Multi-junction solar cells can be implemented in a 2-terminal (2T) and 3-terminal (3T) architecture. In 2T III-V solar cells, the subcells are connected in series by using a tunnel junction that can be grown in the III-V system. However, the currents in the two subcells need to be matched. Alternative designs using a 3T architecture have been proposed [27,28]. These have the benefit of not having to match the currents in the subcells but at the expense of increased interconnection complexity, requiring a third contact at the recombination layer instead of the tunnel junction. The total power output generated by the tandem solar cell is calculated by summing the powers of the two subcells as reported in the taxonomy for 3T tandems proposed by Warren et al. [29].

Figure 1 illustrates the modeled tandem solar cell structure. The bottom cell (Figure 1a) consists of an InP substrate covered with NWs, passivated using ALD-defined ${\mathrm{SiO}}_{x}$, and connected by ITO. A polymer layer, benzocyclobutene (BCB), is deposited in between the NWs, an example of which can be found in Appendix A. The full tandem architecture (Figure 1b) includes, from bottom to top, an electron transport layer (ETL), the perovskite layer (PSK), a hole transport layer (HTL), and the ITO top contact. In the schematic, the metal contacts are indicated to provide the connection scheme, but they are not included in the modeled structure. Device active areas are defined employing a polymer covered with ITO and terminated with a gold pad, providing the top contact according to the schematic provided in Appendix A (backside contact is typically defined by evaporating Ti/Zn/Au on the whole surface).

The optical constants of InP and ITO were obtained from Wallentin et al. [8]. For ${\mathrm{SiO}}_{2}$, the values reported by Gao et al. [30] were used. As an electron transport layer (ETL), ZnO reported by Stelling et al. [31] was selected. For the 3T structure, the perovskite ${\mathrm{CH}}_{3}{\mathrm{NH}}_{3}{\mathrm{PbBr}}_{3}$ reported by Brittman et al. [32] was chosen due to its absorption onset at around 550 nm. However, to achieve current matching in the 2T tandem cell, it was replaced with a lower bandgap perovskite $\mathrm{MAPb}{\left(I_{x}{\mathrm{Br}}_{1-x}\right)}_{3}\;\left(x=0.66\right)$, based on private communication with Tejada [33]. The hole transport layer chosen was PEDOT:PSS reported by Chen et al. [34]. The structure is considered to be surrounded by air.

### 2.2. Numerical Analysis

The bottom cell has a pitch of 500 nm from center to center of the nanowires, with a height of 2 μm and a diameter of 200 nm. Other parameters have been adjusted to minimize parasitic contributions, particularly ITO absorption and reflectivity. Due to computational constraints, the 2D array of nanowires was modeled as a 1D grating. Due to the continuous translation symmetry of the 1D grating, a 2D model can be used instead of the full 3D model necessary for the nanowires. This has been shown to deliver qualitatively similar results [35]. A quantitative device optimization should employ simulations of the full nanorod array. Periodic Floquet boundary conditions are applied in the lateral direction, which allows for a reduction in the computational domain to a single unit cell. In the vertical direction, transparent boundary conditions are applied to allow for outgoing radiation without any unphysical reflections. The illumination is modeled using a plane wave at normal incidence with P polarization; this is due to the homogeneous material distribution seen by the S-polarized light, which does not approximate the nanorod structure.

The optical model was numerically solved using the finite element solver JCMsuite [36]. The mesh was constructed such that the maximum side lengths in each material were less than a quarter of the material dependent wavelength. The basis function polynomial degree was adapted on each element to obtain a goal accuracy of under 0.1%. This typically leads to polynomial degrees in the range of three or four for most elements. The transparent boundary conditions were realized using perfectly matched layers. The reflectance and transmittance into the InP substrate were calculated via integration of the outgoing flux density at the top and bottom computational domain boundaries, respectively. The absorptance in each domain was calculated via the integration of the electromagnetic field absorption density. The spectral response was calculated in the visible range by varying the wavelength from 410 nm to 980 nm in 10 nm steps. The obtained absorptance was used as the upper limit of the external quantum efficiency (EQE) by assuming that all absorbed photons contribute to the obtained current density, effectively setting the internal quantum efficiency of the device to unity. The current density is then given by:(1)$$J_{ph}=-e\int_{410\;\mathrm{nm}}^{980\;\mathrm{nm}}\frac{A\left(\lambda \right)\Phi_{AM1.5g}\left(\lambda \right)}{E(\lambda )}\;d\lambda$$
where *e* is the electron charge, A(λ) is the absorptance, Φ(λ) is the AM1.5g solar spectrum, and E(λ) is the photon energy per wavelength. The current density outside the active layers is calculated to estimate the losses due to parasitic absorption.

## 3. Results and Discussion

### 3.1. Bottom Cell

The structure in Figure 1a is simulated with varying oxide and ITO thickness, and the parameters variation is reported in Figure 1c. The results show that the ${\mathrm{SiO}}_{2}$ thickness does not influence the light propagation and reflectivity of the whole structure, or contribute to the overall parasitic absorption, and thus its thickness is fixed at 25 nm. Conversely, the ITO thickness is a parameter that needs to be adjusted to reduce reflectivity. The ITO thickness on the sides of the NWs is set to be half the vertical thickness of the planar layer in between the NWs to be in line with experimental observations.

In Figure 2a, the absorptance and AM1.5g spectra for different ITO thicknesses are shown. Thinner ITO layers can shift the reflectance peak to a region of lower intensity in the AM1.5g spectrum. This behavior can be attributed to an anti-reflection action as suggested by Chen et al. [37]. The results are in agreement with experimental EQE reported by Wallentin et al. [8]. The simulation results in Figure 2b show that the R spectrum is broader for the 120 nm thick ITO compared to the 100 nm and 80 nm cases. This is due to the thin film interference condition for longitudinal modes. For more information, see Appendix A. An ITO film thickness below 80 nm would be preferable due to the parasitic absorption present in ITO. However, a minimum ITO thickness of 80 nm has been suggested to maintain a low sheet resistance as reported by Kim et al. [38]. In this regard, a higher sheet resistance will have a higher impact on a 3T tandem since, in this configuration, the current is collected at the contact to allow the subcells to work independently, while in 2T tandem, it is used as a recombination layer.

Using the absorptance spectra, the photocurrent density was calculated using Equation (Equation 1) and resulted in 29.4 $\mathrm{mA}/{\mathrm{cm}}^{2}$, 28.8 $\mathrm{mA}/{\mathrm{cm}}^{2}$ and 28.1 $\mathrm{mA}/{\mathrm{cm}}^{2}$ for 80 nm, 100 nm, and 120 nm ITO thickness, respectively. Therefore, the reduction in reflection and parasitic absorption across the visible spectrum for decreased ITO thickness directly corresponds to increased photocurrent density.

### 3.2. 3T Tandem Solar Cell

Once the bottom cell was optimized, the transport layers, contacts and the perovskite were added as shown in the schematic in Figure 1b. To perform the analysis related to the various parasitic contributions. The perovskite overstanding layer, equivalent to the coating layer thickness above the NWs array contact, was kept constant in thickness in the modeling, at 150 nm, close to the minimum absorption length:(2)$$\delta_{p}=\frac{\lambda }{4\pi Im(n)}$$
where Im(n) is the imaginary part of the complex refractive index: $n=\tilde{n}+ik$; thus, $\delta_{p}$ is the minimum thickness required to absorb 64% of the light at a given wavelength.

Varying the electron transport layer thickness from 25 nm to 50 nm does not affect the reflectivity, making ZnO suitable for integration as an ETL in the tandem structure; its thickness is fixed to 35 nm. The variation of the BCB filling in Figure 3 shows how filling the space in between the NWs with perovskite impacts the absorption spectrum and overall photocurrent. It can be observed that by increasing the thickness of the BCB filling and thus reducing the amount of perovskite between the NWs, the total photocurrent remains constant. As shown in Figure 3b, increasing the BCB fill height over 1200 nm consistently changes the ratio in the current generated in the NWs and perovskite; the absorption density change is shown in Appendix A. The equivalent current due to perovskite absorption is ∼0.9 $\mathrm{mA}/{\mathrm{cm}}^{2}$ higher compared to the current generation in the planarized structure, showing the contribution of the absorber in between the NWs. Additionally, the fill height variation simulated for 3T structures can provide an additional parameter for achieving current matching in two-terminal structures.

Figure 4a shows the results of absorptance modeled by varying the perovskite overstanding layers thickness, ranging from 225 nm to 950 nm, on top of InP nanowires (NWs), and with the BCB fill height set at 1200 nm. This choice takes into account both the results obtained from the current optimization previously shown and the challenges in depositing perovskite in between the NWs. Figure 4b shows the photocurrent variation, increasing the perovskite overstanding layer thickness. The total photocurrent does not change significantly, despite changes in the photocurrent ratios generated in the NWs and perovskite layers; the decreasing trend can be attributed to increased reflection at the interface.

### 3.3. 2T Tandem Solar Cell

As an alternative approach, we modeled a two-terminal structure that allows for the summation of open-circuit voltages ($V_{\mathrm{oc}}$) at the expense of current-matching requirements; this structure is simpler in terms of fabrication and wiring. In this case, perovskite used in the simulations was chosen to be $\mathrm{MAPb}{\left(I_{x}{\mathrm{Br}}_{1-x}\right)}_{3}\;\left(x=0.66\right)$ with a 1.8 eV bandgap [39], which is optimal for obtaining current matching with InP, whose bandgap is 1.34 eV. The intermediate ITO layer is excluded from the modeled structure, considering the feasibility of III-V tunnel diodes with similar refractive index and small thickness [40].

In this case, the current matching condition results in ∼14.1 $\mathrm{mA}/{\mathrm{cm}}^{2}$, calculated for wavelengths between 280 nm and 980 nm. Assuming a $V_{\mathrm{oc}}$ of 0.9 V for the bottom cell (Wallentin et al. [8], sample E) and $V_{\mathrm{oc}}=E_{\mathrm{gap}}-0.3V$ for the perovskite top cell, along with a fill factor of 80%, the theoretical power conversion efficiency is 26.7%. The theoretical power conversion efficiency, comparable to the one reported by Raja et al. [23], is limited by the available material optical constants, indeed considering that a lower bandgap bottom cell and slightly wider top cell could lead to a better spectrum splitting, thus increasing the short circuit current. However, with this structure, we want to show a structural scheme of easy integration with minimal material consumption and relying on the NWs anti-reflective effect. In fact, other III-V semiconductors can provide a lower bandgap for the bottom cell, such as InAsP.

The 2T solar cell based on the planar structure, consisting only of the corresponding materials stacked (schematic available in Appendix A), has been compared to the one based on the NWs solar cell in figure. The corresponding absorption profiles are shown in Figure 5a,b. In Appendix A, the absorption density is shown. We observe that the reflection contribution of the planar structure is higher; for both structures, the absorption below the bandgap is neglected. In fact, the minimum current generated in the bottom cell of the planar structure is approximately 9.5 $\mathrm{mA}/{\mathrm{cm}}^{2}$, while in the nanostructured tandem solar cell, it is significantly higher, at 14.1 $\mathrm{mA}/{\mathrm{cm}}^{2}$. Moreover, considering the amount of current that is lost due to reflection in the planar case, it accounts for approximately 7.1 $\mathrm{mA}/{\mathrm{cm}}^{2}$, whereas for the nanostructured tandem solar cell, it is significantly lower, at approximately 3.5 $\mathrm{mA}/{\mathrm{cm}}^{2}$. The effect of nanostructures is comparable to the results reported by Sahli et. al. [18] in perovskite–silicon tandem solar cells, demonstrating that reflection losses could be reduced to below 2 $\mathrm{mA}/{\mathrm{cm}}^{2}$ by optimizing the perovskite deposition method to enable conformal coating of perovskite on nanostructured surfaces. These results demonstrate the benefits of nanostructures to trap light more efficiently, reduce reflection and thereby enhance the performance of the tandem solar cell.

## 4. Conclusions

In this manuscript, we investigate the optical response of a tandem architecture, consisting of perovskite as the top cell and InP NWs as the bottom cell.

The results highlight the impact of ITO on parasitic absorption and the necessity of tuning its thickness. Simulations of the 3T structure demonstrate that the space between the NWs can be used to harvest a small portion of light by perovskite, and the ratio between the current generated in the perovskite and the III-V solar cell can be adjusted. Competitive absorption between InP NWs and perovskite does not lead to an increase in the output current; however, this architecture allows for decoupling the operation of the two subcells. The optimization performed on the 2T solar cell reveals the benefits of employing nanostructures for the bottom cell. Using NWs results in a reduction in reflection compared to the planar counterpart, allowing for improved light absorption using a fraction of the material needed for a film covering 100% of the surface. Nevertheless, both III-V semiconductors and perovskites offer bandgap tunability and spectrum optimization for the two subcells, thereby offering the potential for further developments. The transport layers represent a challenge for absorption enhancement, due to affecting the reflection at the front interface; moreover, they exhibit high parasitic absorption. Different transport layer materials should be investigated to improve the overall optical performance of the tandem solar cell structure.

## Figures and Tables

**Figure 1 nanomaterials-14-00518-f001:**
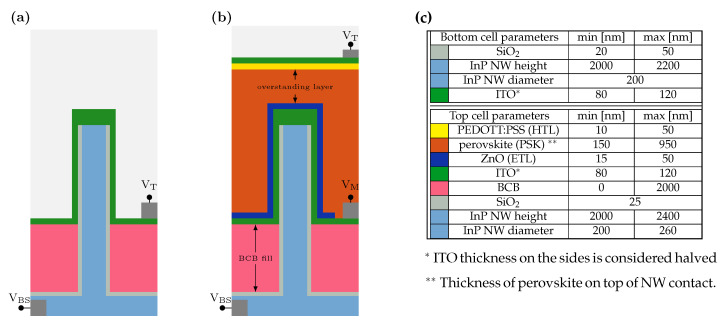
(**a**) InP NW solar cell. Corresponding materials are reported in table (**c**). (**b**) Starting from the bottom cell, transport layers and perovskite for the tandem structure. For both structures, the front contact is ITO, and the back contact is made by Ti/Zn/Au at the backside of the substrate. $V_{T}$ indicates the top contact, $V_{M}$ the mid contact for the 3T solar cell, and $V_{\mathrm{BS}}$ the backside contact. (**c**) Minimum and maximum values of thicknesses used in this study.

**Figure 2 nanomaterials-14-00518-f002:**
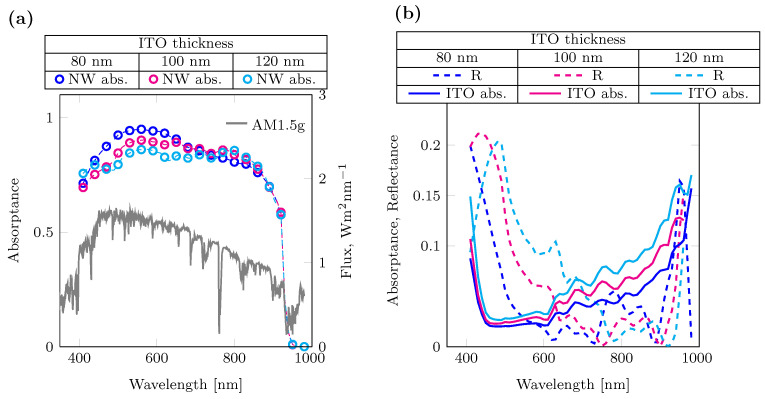
Bottom cell optimization. (**a**) Calculated absorptance of the bottom InP subcell for different ITO thicknesses (dotted lines), the AM1.5g spectrum is reported (solid gray line). (**b**) Reflectance (dashed lines) and parasitics contributions for different ITO thicknesses (solid lines).

**Figure 3 nanomaterials-14-00518-f003:**
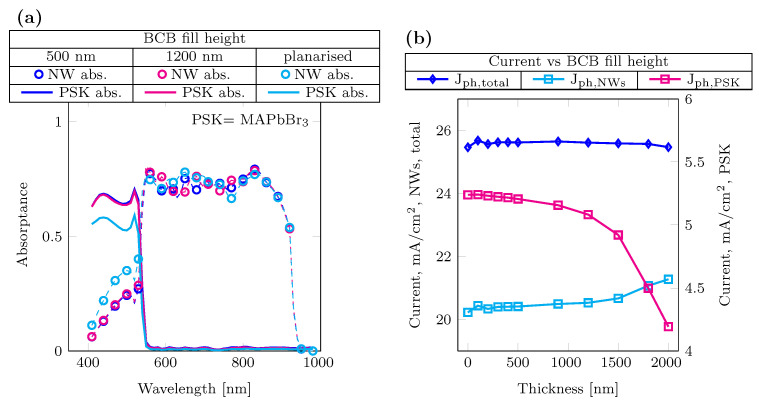
BCB fill height optimization (**a**) absorptance profiles for different BCB thicknesses 500 nm (dotted blue line), 1200 nm in (dotted magenta line) and planarized in (dotted cyan line), corresponding to a fill height of 1975 nm. (**b**) Current behavior for different BCB fill heights. NWs and total photocurrents (diamond blue marker and cyan square marker) refer to the left y-axis, and perovskite photocurrent (magenta square marker) to the right y-axis.

**Figure 4 nanomaterials-14-00518-f004:**
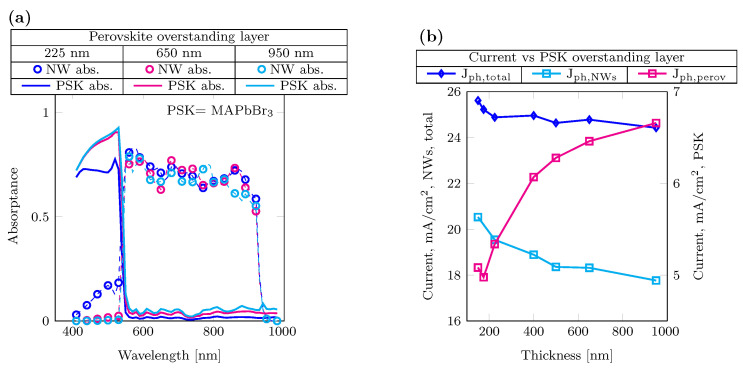
Variation of perovskite thickness. (**a**) Absorptance for different perovskite overstanding layer thicknesses on top of the NWs: 225 nm in blue, 650 nm in magenta and 950 nm in cyan. (**b**) Current behavior for different perovskite overstanding layer thicknesses. NWs and total photocurrents (diamond blue marker and cyan square marker) refer to the left y-axis, and perovskite photocurrent (magenta square marker) to the right y-axis.

**Figure 5 nanomaterials-14-00518-f005:**
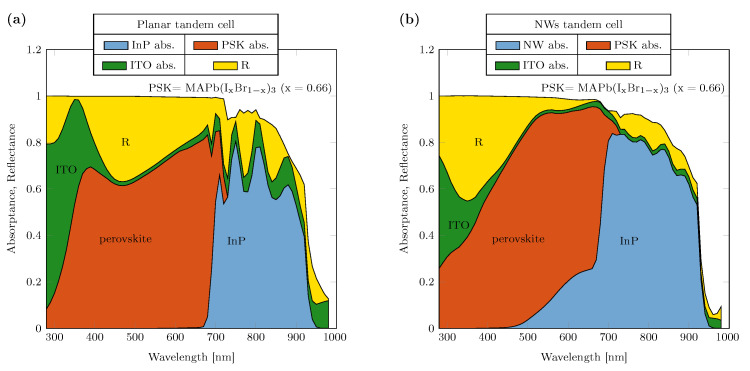
Absorptance of 2T devices for (**a**) planar tandem architecture. The perovskite and InP thicknesses are respectively 1500 nm and 2000 nm. (**b**) NWs tandem architecture. Perovskite overstanding layer thickness = 250 nm, InP NWs diameter = 240 nm length = 2200 nm.

## Data Availability

Data underlying the results presented in this paper are not publicly available at this time but may be obtained from the authors upon reasonable request.

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
