# Peer review of "Optical Analysis of Perovskite III-V Nanowires Interpenetrated Tandem Solar Cells"

_nanomaterials, 2024, doi:10.3390/nano14060518_

Round 1

Reviewer 1 Report

Comments and Suggestions for Authors

See attachement

Author Response

Reviewer 1:

  • Is it possible to also compare with the light reflection loss in a more common perovskite-Si tandem cell?

Light reflection losses can be compared with the available literature. Considering for example the structure proposed by Chen et. al. (10.1117/1.JPE.8.022601) where structuring the Si front side reduced reflection losses from 5.5 to 2.8 mA/cm2 equivalent current density. A more interesting system is the one reported by Sahli et. al. (10.1038/s41563-018-0115-4), consisting of a perovskite-silicon tandem solar cell nanotextured on both sides. In this case reflection losses could be reduced to below 2 mA/cm2 by adaption of the perovskite deposition method enabling conformal coating of perovskite on pyramidal textures. This consideration has been introduced in text in the paragraph discussing the results obtained for the 2-terminal structure. (L220)

Reviewer 2 Report

Comments and Suggestions for Authors

Metal halide perovskites (MHPs) have attracted growing interest in the field of photovoltaics due to the defect tolerance and bandgap tunability. In this paper, Tirrito et al. investigated the integration of MHPs with III-V NWs solar cells in both 2-terminal and 3-terminal configurations. Both materials allow for bandgap tunability in the range of 1.2 eV to 2.4 eV, making them ideal for use in a tandem structure. Using NWs results in a reduction in reflection compared to the planar counterpart, allowing for improved light absorption using a fraction of the material needed for a film covering 100% of the surface. In all, this paper provides significant insights into NW tandem solar cell optics, showing their potential in improving photovoltaic performance. Therefore, I would recommend this manuscript for publication.

Comments on the Quality of English Language

Minor editing of English language required.

Author Response

The language has been improved in some areas of text to provide more clarity, such as to describe the schematic. (L89), while in L174 and L179, the grammar has been corrected. A relevant work from Sahli et al. has been referenced to better explain the results. (L220).

Reviewer 3 Report

Comments and Suggestions for Authors

In this paper, the optical behavior of tandem solar cells, composed of perovskite and InP nanowires, are modeled. Some structural parameters are optimized, and several structural schemes considered with the aim of minimizing material consumption and simplifying integration. Overall, the paper is clearly written and scientifically sound. The concept of bringing perovskite and nanowire solar cells together in a tandem structure is interesting.  Still, very much can be found in the literature about perovskite and nanowire solar cells as well as tandem structures. The author should express more clearly what exactly is new in their work. Also, a comparison of the light efficiency with other recent tandem cells would have been useful and should be included.

Additional minor comments:

Spell out “BCB” (benzocyclobutane?) where it first appears.

Apparent inconsistency in the colors in Fig. 1. In Fig. 1b, there is a yellow layer on top of the PSK which should present the HTL. In Fig. 2c, the HTL layer is labelled with a brownish color.

L49. Take out the word “more”.

Author Response

  • In this work we consider for the first time III/V nanowires and perovskite integration, moreover, alternatively, for example, to the 4-terminal structure proposed by Raja et. al., we focused on 3-terminal and 2-terminal structure. They obtained a power conversion efficiency of 27%, comparable to our 2-terminal structure, however, the processability of their structure is more challenging, involving perovskite NWs and a 4-terminal structure, which involves a more complex connection scheme. Moreover, our structure is extremely compact leading to limited material consumption. This consideration has been added to introduction of the paper. (L68)
  • Spell out “BCB” (benzocyclobutane?) where it first appears.

This comment has been addressed in the text

  • Apparent inconsistency in the colors in Fig. 1. In Fig. 1b, there is a yellow layer on top of the PSK which should present the HTL. In Fig. 2c, the HTL layer is labelled with a brownish color.

This comment has been addressed in the text.

  • Take out the word “more”.

I could not  find word more in L49, I think is in L149. However, this comment has been addressed in the text.